# Cancer patients' experiences of the diagnosis and treatment of incidental pulmonary embolism (a qualitative study)

Naima E. Benelhaj[1]*, Ann Hutchinson[2], Anthony Maraveyas[3], Miriam J. Johnson[2]

1 Hull York Medical School, University of Hull, Hull, United Kingdom, 2 Wolfson Palliative Care Research Centre, Hull York Medical School, University of Hull, Hull, United Kingdom, 3 Queen's Centre for Oncology and Haematology, Hull and East Yorkshire Hospitals NHS Trust, Hull, United Kingdom

☉ These authors contributed equally to this work.

* naoma_73@yahoo.co.uk

**Data Availability Statement:** All relevant data are within the paper.

**Funding:** The author(s) received no specific funding for this work.

## Abstract

### Background

The diagnosis of symptomatic cancer-associated thrombosis often causes distress and alarm for patients, especially for those unaware of the risk, or the signs and symptoms to look out for. There are few data about cancer patients' experiences of incidentally diagnosed pulmonary embolism (IPE), where lack of warning (recognised signs, symptoms) may cause delayed diagnosis and aggravate distress.

### Objectives

To explore cancer patients' experience of the diagnosis of and living with incidental pulmonary embolism treated with anticoagulation.

### Methods

A qualitative study using modified grounded theory approach. Semi-structured interviews were conducted as part of a mixed- methods prospective observational survey study of consenting patients with IPE. Data were subjected to thematic analysis. The qualitative findings are presented.

### Findings

Eleven participants were interviewed (mean age 68.3 years, range 38–82 years; various forms of cancer and stages). Three major themes and one cross-cutting theme were generated. Theme (1): IPE is experienced in the context of cancer and concomitant comorbidities. Issues are understood in the shadow of–and often overshadowed by—current serious illness. Theme (2): Being diagnosed with IPE. Misattribution to cancer or other comorbidities caused delay in help-seeking and diagnosis. Theme (3): Coping with anticoagulation. Participants' incorporated anticoagulation treatment and its effects into their daily routine with acceptance and stoicism. Finally, the cross-cutting theme relates to a lack of information and uncertainty, contributing to distress throughout the experience.

**Competing interests:** The authors have declared that no competing interests exist.

## Conclusion

The diagnosis of IPE was upsetting and unexpected. Expert and timely information was valued by those with IPE. Education called for about the increased risk of cancer-associated thrombosis and the signs and symptoms to be aware of.

## Introduction

Incidental pulmonary embolism (IPE) is defined as a filling defect of one or more pulmonary arteries seen on imaging ordered for indications other than suspected PE [1]. The incidence of IPE is estimated to be 3.6% in cancer computed tomography (CT) scans [2] and IPE accounts for over one-third of cancer-associated PE [3]. Cancer-associated IPE carries a substantial risk of recurrent venous thromboembolism and mortality, despite anticoagulant treatment [3–5] and the survival rate of cancer patients with IPE is poorer than those without [5–8]. Recent guidelines recommend the same treatment for both IPE and symptomatic PE [9]. The psychological impact of cancer-associated thrombosis (CAT) has been described previously [10, 11]. Patients, and sometimes their clinicians, are unaware of their risk of CAT [12–15]. Therefore they do not recognise the signs or symptoms of CAT which often mimic those of the underlying cancer and are therefore misattributed to the cancer [11, 12]. For some, the legacy of this experience is living with anxiety about a further, potentially fatal, thromboembolism [16]. Emerging evidence indicates that this may be even worse for people with IPE compared to those being in a process of being investigated for symptoms of CAT that entails some forewarning. We therefore explored the experience of diagnosis and anticoagulation in people with cancer-associated IPE.

## Methods

### Study design

This qualitative study used semi-structured interviews and was part of a prospective observational survey study. We present the qualitative findings.

### Ethics approval

Ethical approval was granted by the East Midlands—Nottingham Research Ethics Committee and HRA (REC ref: 16/EM/0474) (IRAS project ID: 216188) approved 21/12/2016. The procedures used in this study adhere to the tenets of the Declaration of Helsinki. All participants provided written consent, which was taken by the researcher prior to the interview commencing.

### Methodology

A modified grounded theory informed approach was chosen [17, 18] as although there were few data regarding experience of IPE—the phenomenon, [19] there were for non-incidentally diagnosed CAT. We sought to explore and describe cancer patients' experiences of diagnosis of and living with IPE.

Semi-structured interviews [20] used a topic guide developed through the team's experience, and published literature, and underpinned by Uncertainty Theory [21]. The questions were open-ended and flexible to ensure that the same issues were explored with all participants

whilst allowing individuals to raise unanticipated issues to be explored in subsequent interviews.

## Setting/Recruitment

Participants were recruited from a nurse-led service pathway for patients with cancer diagnosed with IPE [22] at a single tertiary cancer centre (February to September 2017). This was a convenience sample of consenting participants completing an observational survey study. Thirteen potential patients were approached over the phone by NB and informed consent was obtained.

## Consent to participate and consent for publication

Informed consent was obtained from all individual participants included in the study, including publishing part of their quotes with no identifiable data.

## Data collection

Individual face-to-face semi-structured interviews were conducted within one month of the diagnosis of IPE by NB who was not part of the clinical team, had no prior relationship with participants and no access to their medical records. The interviews were conducted within the regular follow up at the hospital or at participants' homes according to the patient's preference and lasted for 15–30 minutes. Where carers were present and willing to participate, dyad interviews were conducted. The interviews were focussed and short in order to minimise the participants' burden. However, those who wished to talk for longer were able to, and breaks were offered as required. The topic guide included questions about participants' experience of: i) life before the diagnosis of an IPE, ii) being diagnosed with an IPE, iii) anticoagulation in the context of incidental diagnosis, iv) the effect of the IPE on their life and their cancer journey, v) everyday life living with cancer and an IPE. Interviews were carried out until theoretical saturation was achieved [23].

## Reporting

This article is reported in accordance with the Consolidated criteria for reporting qualitative research (COREQ) checklist [24].

## Analysis

Transcripts were audio-recorded and transcribed verbatim. Thematic analysis [25] was chosen to analyse the study results through the lens of Uncertainty Theory [21], driven by a grounded theory approach [26].

The analysis involved five phases. First, familiarization with the data by listening to the recordings, reading and re-reading all the interview transcripts, then NB and MJ independently conducted a line by line coding of two transcripts. Following discussion (NB, MJ), a code book was formed and used by NB to code all the transcriptions, who remained alert to possible new codes.

Codes were then discussed (NB, MJ) and an initial thematic map generated. Finally, defined themes were then formed to "tell the story" of the participants' experience. Carer and patient data were analysed together.

## Findings

Thirteen eligible participants were invited; 11 participants consented (4 women; mean 68.3 years; range 38–80 years), two declined without giving any reason. The most common primary tumour sites were gastrointestinal cancer (n = 5), lung cancer (n = 3) and genitourinary cancer (n = 3). Ten had advanced stage cancer. ECOG status: grade 0 = 5; 1 = 4, 2 = 1, 1 = unknown). Two were interviewed with their carers. All participants were white British.

## Themes

Three major themes, and a cross-cutting theme, were generated from the data. These are reported below with illustrative quotes (see also Table 1).

**Theme one: IPE in the context of cancer and concomitant comorbidities.** Participants described living with cancer as a journey of ups and downs, hopes and disappointments

*"I thought cancer was getting better, till I was told that cancer was growing again. It was quite high, quite aggressive and it came down by hormone treatments." (P02)*

Despite this, participants tried to live as normally as they could. Many described ways to overcome the side effects of cancer treatment, and other restrictions where possible, even when extensive.

*"You know, I have to be strong both for my husband and for myself, you know it is not easy-...it is not when we were first told, we have tears, and hugged and kissed and all thoughts... but, we have to get on with it." (P04)*

**Table 1. Themes and subthemes summary.**

| Themes | Subthemes | Exemplar Quotation |
|---|---|---|
| IPE in the context of cancer and concomitant comorbidities | Life with cancer | *"Well it was upsetting to find out it was growing again" (P02)*<br>*"There is always ups and down but the chemo knocked me out" (P08)* |
| | IPE in context of cancer | *"If I get a pain I immediately related it to cancer, if I am a bit fluctuant or wind in my stomach I put it down to cancer" (P01)*<br>*"I have heard of it, but I did not know it is associated with cancer treatment". (P02)* |
| | Symptoms misattribution | *"I am asthmatic (but under control) I have felt short of breath like going upstairs" (P02)* |
| Being diagnosed with IPE | Diagnostic process | *"They phoned me up and said there are a small blood clots in your lungs, would you please come in and we will deal with it." (P03)* |
| | | *I was waiting for them to come and take the needle out of me, and the lady came over and asked me would I go with her to a little room and there was a doctor already, he explained that they this blood clot and he would be leaving a massage for one of the specialist nurses who should ring me back the following day." (P04)* |
| | Response to IPE diagnosis | *""It was a big shock. Well, I think I was such a chock I did not really take things in" (P02)* |
| | Need for information | *"So, I did what you shouldn't do and as soon as I get home I went on Google and I found out all about the symptoms and what it could be" (P07)* |
| Coping with anticoagulation | Acceptance | *"It has to be done isn't" (P03)* |
| | Side effects | *"I am all bruised all around here (abdomen), it is black and blue, the whole of my stomach, but that dose not affect me" (P05)* |
| Cross-cutting theme: Lack of information and uncertainty | | *"not blaming the staff, I think I've should have more information, but it is not there if you know what I mean, it is not" (P01)*<br>*"I mean, it is going to be a long while if the blood clot will ever disappear, which is worry for life really" (P01)*<br>*"Each time you think Why me? Is it something I Have done wrong, or is it my life style" (P02)* |

Some were stable and pleased they were doing well. For those participants, an unsuspected complication was unexpected, but they took it in their stride, pleased that the cancer was responding to treatment.

*"I was also over of the moon because he told me that my cancer has shrunk by 50% so I wasn't bothered by the clot, I was so happy that the cancer has shrunk." (P10)*

For most participants the diagnosis of IPE was co-incident with the diagnosis of cancer or soon after, at the start of treatment. This added a significant burden on them at an already difficult time and seemed to contribute to a delay in seeking help.

*"After I had the first session of the chemotherapy, I got the severe chest pain, I didn't do anything about it for a week, I wasn't, I should have done probably." (P05)*

Although the diagnosis of pulmonary embolism was an incidental finding discovered on a CT scan ordered for cancer staging or follow up, participants reported having thrombosis (PE/DVT) related symptoms even though these had not been recognised These new or worsening symptoms were ignored by participants relating them to side effects of cancer, its treatment or to other comorbidities leading to delays in seeking any medical help.

*". . .and I do occasionally feel out of breath and I have used my (inhaler) I am asthmatic (but under control) I have felt short of breath like going upstairs, . . .even now I feel little out of breath, but is that the clots?" (P02)*

Health care professionals also misattributed their symptoms. That not only delayed diagnosis of PE but put participants at risk of inappropriate medication for incorrect diagnoses.

*"I woke up in the morning with top of my calf red, a bit sore and feeling hot. . . she [doctor] said oh no I don't think it is that [CAT], she said it is more likely cellulitis, so put me on a course of antibiotics but then of course 2 or 3 weeks. . ." (P06)*

*"After the first session of chemotherapy, I got really bad chest pain, so I was referred to the cardiology department to check on my heart. So the chemotherapy continued and every session get progressively worse." (P 05)*

Eventually, P05 went to the GP who arranged a CT scan.

*"So, when my last session of my chemotherapy was finished, the GP . . . arranged for me to have a scan." (P05)*

Others had no, or minimal, new/worsening symptoms and thus had no warning of a new problem with their cancer or its complications

*"I have no pain. No symptoms at all, nothing whatsoever." (P01)*

## Theme two: Being diagnosed with IPE

Some received their diagnosis on the same day whilst still in hospital for their imaging, while others were told over the phone after they went home. Those who had their diagnosis on the same day appreciated being able to talk to a member of staff who referred them to the IPE service.

"Yes, he was *very* good, . . . the radiographer was waiting in the room and he explained everything for me. And the following day it all went as he said would (sic)." (P04)

Those who received their diagnosis over the phone after they went home, described it as an inappropriate way of breaking bad news. They felt rushed with instruction to come the next day to start treatment with insufficient information which made them more worried.

*"Which was a bit worrying because, being rushed when you got cancer you would say Oh what else?" (P01)*

*"So I was still waiting to find out results of the cancer search, and then to find out I got this. It was in a big shock (sic). It came out of blue. It was quite a shock. It was a big shock." (P02)*

Issues relating to information are reported in the cross-cutting theme below.
Some participants, especially those who were truly asymptomatic, seemed less bothered by this new finding taking the lack of symptoms as a sign that it was not serious.

*"He said you know this does happen, with the cancer patients they formed the blood clot, it is a normal thing, but if it is normal, nothing to worry about it." (P10)*

### Theme three: Coping with anticoagulation

All participants started on low molecular-weight heparin (LMWH) and continued for six months. One patient transferred to Direct Oral Anticoagulants (DOACs) due to a drug interaction.
While some expressed preference to tablets instead of injections, participants considered the use of LMWH injection acceptable although unpleasant to do for months or longer.

*"Tablets are easy, aren't they? A bit of water and down that is it." (07)*

*"You know six months, 180 days, 180 times I've got to have this, and again you will adapt to these things."(P10)*

*"The lady doctor diagnosed me said Oh you might have to inject for life, which was a big shock for me." (P02)*

One patient highlighted the difficulties of doing the injection by himself due to concomitant comorbidity.

*". . .but if my hands won't shake I would have no worries what so ever." (P01)*

Some found incorporating this alongside other cancer related issues a logistical challenge.

*"I am on hormone Zoladex that is why I am trying to leave a space on my stomach." (P02)*

*"She got a colostomy bag, and she got bruises, few ones here, and we did one in her leg." (P08)*

Bruising was a common side effect of LMWH; acceptable for most. Participants were aware of the risk of bleeding but expressed no concerns other than of needing to be more careful.

*"Just need to be careful, if I am going anywhere, and I wear gloves if I work in the garden." (P03)*

Adherence was a burden for some participants; remembering it, taking treatments with them and finding somewhere to do the injection if they were out.

*"Just remembering to do it some days, and some days else you say Oh well. . . that is the only thing. And I suppose if I go away or anywhere I have to take a supply with me." (P05)*

Overall, participants found ways to cope with the new situation.

*". . .being negative, because does not do any good, just making you feel worse, just got to be positive all the time, I was fine by it, I was fine." (P08)*

## Cross-cutting theme: Lack of information and uncertainty

The overwhelming majority of participants reported lack of knowledge about the association between cancer, cancer treatment and the risk of thrombosis.

Participants were well aware and supported about the side-effects of chemotherapy and knew how to reduce the effects, but despite the study setting being in a hospital with CAT/IPE clinic, none remembered being informed about the risk of CAT, or the symptoms and signs to watch for prior to their diagnosis of IPE.

*"No. It would be nice if I've been told. I mean I know there are the normal side effects of having chemotherapy, you know tiredness, whatever of loss of appetite, and bad mouth and that sort of things, but nothing about the blood clot."(P05)*

This lack of knowledge contributed to the delays in diagnosis when participants misattributed their symptoms to cancer, its treatment, or other medical conditions.

Unless participants were seen immediately in the specialist IPE clinic, when information was provided it was unclear and insufficient. Participants therefore sought information from other sources, such as the internet, resulting in some receiving alarming information in an unsupported way and sometimes conflicting with professional advice already received.

*"So I did what you shouldn't do and as soon as I get home I went on Google and I found out all about the symptoms and what it could be." (P07)*

Others were left feeling it was their fault for not being more proactive in seeking information in clinic.

*"I think, some of it is my fault. I've should have asked, if I asked it would be there. . .I'm not blaming the staff, but think I've should have more information, but it is not there if you know what I mean (sic)." (P01)*

When the diagnosis of IPE was accompanied by poor information, this increased the level of uncertainty: why they (in particular) had developed this complication, what was the likely clinical course of IPE, what if they developed further blood clot and how would they know if it did, and would the anticoagulation treat the IPE successfully.

*"I asked how I will find out if the clot been reduced. And the only way that they can tell me is by a scan which is in three months' time, so one worry is what happen in during three months' time? Is it going up or down? I also worried about you suddenly got a pain." (P01)*

*". . .and each time you think Why me?" (P02)*

*"When you think emboli! What if it then moves? My mind was not happy." (P03)*

The lack of immediate information, answers to questions, and support aggravated the distress in response to the diagnosis of IPE.

*"I didn't know how severe having a blood clot, I mean no one said, they did not say oh a blood clot in your lung or in the bottom of both lungs. Maybe that would worried me more if they said that, that is why they don't." (P05)*

*"Upset, I wanted to speak to a consultant in here, ideally you want to speak to Dr XX, to get some advice." (P07)*

Participants appreciated the specialist IPE clinic where they were able to access accurate, detailed information and have the chance to talk to an experienced specialist nurse who explained the situation to them.

## Discussion

The cancer and other significant comorbidities shaped the context in which the participants experienced the diagnosis of IPE. For those with signs and symptoms of PE, these had been misattributed by both patient (delay in help-seeking) and clinician (investigation for PE not initiated) and at times wrongly treated by clinicians. The diagnosis of IPE for many was therefore a shock, with distress at the delay in diagnosing a potentially life-threatening complication. The incidental diagnosis meant that patients had no forewarning unlike those who knew PE was a possibility. Those without symptoms seemed to fare better psychologically, assuming "asymptomatic" to be "less dangerous". For some, the relief of the cancer response shown by CT outweighed worry about the IPE. Lack of information was a major concern regarding; i) the risk of thrombosis, the signs and symptoms to watch for, how and when to access help, ii) the diagnosis of IPE, lack of timely information about the impact and treatment of IPE, and iii) how best to manage IPE treatment alongside the effects of and treatment for their cancer and other conditions. The specialist clinic was valued as providing expert information and professional support which reduced the uncertainty generated by this new occurrence.

Diagnostic overshadowing was first described for people with mental illness, where symptoms of physical illness were misattributed to their mental health state, leading in some cases to increased mortality [27, 28]. Diagnostic overshadowing by a cancer diagnosis [11] causing misattribution or missing of symptoms and signs of thrombosis is now a consistent finding in a number of CAT studies [10–16, 29, 30] leading to delays in seeking medical help, diagnosis, treatments, and increased patient burden. In this study some patients felt that the thrombosis —seen as potentially fatal—was worse than the cancer. Interestingly, French researchers found no such anxiety, although they did find poorer anticoagulation adherence [14]. This may have been because the seriousness of thrombosis was not realised, perhaps as a consequence of a more physician-led clinical decision making style [14]. Despite calls for routine education for patients regarding the risk of thrombosis [11] and a recent trial of a patient-education video reducing presentation delays from a mean 8.9 to 2.9 days supporting the use of education materials in this way, this still does not happen in practice [31]. Further, it brings into question oncologists' training, and suggests a need for a lower threshold to consider thrombosis as an explanation for new or worsening signs and symptoms.

In this study, participants not only were unaware of the risk of thrombosis, but also did not know of the possibility of incidentally diagnosed venous thromboembolism from routine

cancer care scans [12, 29]. At the time of interview, all our participants had a defined treatment and follow-up plan under the care of a specialist IPE clinic [22]. However, not all received accurate, timely and tailored information quickly. Receiving the news of IPE by phone, with an urgent call to clinic added to their distress. Face to face discussion of these issues in a specialist clinic was positively viewed, but otherwise led to searching for information through unreliable sources which could feed their anxiety. This finding supports recent calls for the same sensitive approach needed for any breaking bad news scenario [13, 30], and the need for person-centred information to redress uncertainty-related anxiety [21]. In our study, those with symptoms were more distressed and fearful consistent with post-traumatic distress from cancer-associated PE being related to perceived seriousness of the health threat [16, 32] this in the absence of the knowledge that their symptoms may also have adverse prognostic significance [33].

As with most studies in countries with a culture of joint doctor-patient clinical decision making, self (or nurse)-injection of LMWH was accepted and incorporated into their established cancer care routine, although this could be challenging e.g., with an -ostomy, hormonal injections. Again, we found that although tablets were preferable, an injection was acceptable if this was effective [15, 34, 35]. However, participants had no prior warning and had to adjust to this more quickly than if they were aware that injections might be needed.

The uncertainty inherent in both incidental diagnosis of pulmonary embolism and long-term treatments again brought the importance of information into focus.

### Strengths and limitations

When we conducted this study, it was the first to specifically explore experiences of cancer patients living with incidental-PE and still adds new data to previously reported experiences of patients with CAT.

The limitations are those inherent with all qualitative work; the findings are not and are not intended to be generalisable. Participants were recruited from a specialist cancer-related IPE out-patient nurse-led service and may represent care unavailable elsewhere. The main limitation is the small sample size, but no new codes arose from latter interviews, suggesting data saturation was achieved.

### Implications

Information must be carefully delivered given the "out of the blue" nature of the diagnosis, diagnostic overshadowing and uncertainty. In our previous systematic literature review [11] we called for raising awareness and patient education about CAT. A recent study showed that an educational material (video) for patients as part of routine care led to earlier reporting of CAT, and chemotherapy nurse education led to changes in their perception and prioritization of potentially related symptoms [30]. Thrombosis education should be implemented routinely as part of the cancer treatment planning, in the same way as patients are educated about neutropenic sepsis and spinal cord compression. Diagnostic overshadowing is pertinent to patients with IPE; clinicians should be wary and have a low threshold for considering CAT.

Those diagnosed with thrombosis need timely and tailored information from trusted sources. Clinicians should acknowledge uncertainty-related anxiety caused by IPE and CAT in general.

### Conclusion

Like people with CAT, those with IPE were distressed by the diagnosis, which was often delayed, with symptoms often missed or misattributed, "overshadowed" by their cancer or

comorbidities. Distress was aggravated by the lack of warning, even if, in retrospect they had symptoms. Timely, face-to-face and expert tailored information was greatly appreciated and helped ameliorate uncertainty-related distress. Routine education for patients and clinicians should be provided.

## Acknowledgments

The authors thank the participants who kindly shared their insights with us. This work would have not been successful without the following individuals for their generous time, space, and support of this project: The nurses staff at the IPE service at the Queen's Centre for Oncology and Haematology, Castle Hill Hospital, UK, Hull and East Yorkshire Hospitals NHS Trust; Lyn Harrison (Clinical Research Data Manager), Hilary Clarke (Oncology Data Manager); Kathryn Date (Research Development Assistant) and the Macmillan cancer support staff.

## Author Contributions

**Conceptualization:** Naima E. Benelhaj, Anthony Maraveyas, Miriam J. Johnson.

**Formal analysis:** Naima E. Benelhaj.

**Investigation:** Naima E. Benelhaj.

**Methodology:** Naima E. Benelhaj, Anthony Maraveyas, Miriam J. Johnson.

**Supervision:** Anthony Maraveyas, Miriam J. Johnson.

**Validation:** Miriam J. Johnson.

**Visualization:** Anthony Maraveyas, Miriam J. Johnson.

**Writing – original draft:** Naima E. Benelhaj.

**Writing – review & editing:** Naima E. Benelhaj, Ann Hutchinson, Anthony Maraveyas, Miriam J. Johnson.

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
