## [Decision Letter · Decision Letter 0]

10 Aug 2022

PONE-D-22-00962

Cancer patients’ experiences of the diagnosis and treatment of incidental pulmonary embolism

PLOS ONE

Dear Dr. Ben Elhaj,

Thank you for submitting your manuscript to PLOS ONE. After careful consideration, we feel that it has merit but does not fully meet PLOS ONE’s publication criteria as it currently stands. Therefore, we invite you to submit a revised version of the manuscript that addresses the points raised during the review process.

Please note that we have only been able to secure a single reviewer to assess your manuscript. We are issuing a decision on your manuscript at this point to prevent further delays in the evaluation of your manuscript. Please be aware that the editor who handles your revised manuscript might find it necessary to invite additional reviewers to assess this work once the revised manuscript is submitted. However, we will aim to proceed on the basis of this single review if possible. 

We look forward to receiving your revised manuscript.

Kind regards,

Vanessa Carels

Staff Editor

PLOS ONE

https://journals.plos.org/plosone/s/file?id=ba62/PLOSOne_formatting_sample_title_authors_affiliations.pdf".

Reviewers' comments:

Reviewer's Responses to Questions

**Comments to the Author**

1. Is the manuscript technically sound, and do the data support the conclusions?

Reviewer #1: Yes

2. Has the statistical analysis been performed appropriately and rigorously? 

Reviewer #1: N/A

3. Have the authors made all data underlying the findings in their manuscript fully available?

Reviewer #1: Yes

4. Is the manuscript presented in an intelligible fashion and written in standard English?

Reviewer #1: Yes

5. Review Comments to the Author

Reviewer #1: An interesting piece of qualitative research, which adds to the current body of literature. The article is well structured, appropriate methodology, outcomes supported by appropriate quotes. On the whole the article is easy to read and follow. I do however have a few suggestions questions which I hope once addressed will make the artcile easier to follow.

Paragraph below (page 8) does not read well, difficult to make sense of it.

137 Although the diagnosis of pulmonary embolism was an incidental finding discovered on a CT scan ordered for cancer staging or follow up, participants reported having thrombosis (PE/DVT) related symptoms even though these had not been recognised where carers were present and willing to participate, dyad interviews were conducted.

Re quote below – this may be correct but need to check did the respondent say ‘wrap gloves’

198 “Just need to be careful, if I am going anywhere, and I wrap gloves if I work in the garden.”

What information do you present to support this statement below - in capturing information on the 11 patients interviewed did you ask about symptoms of PE? To enable you to state those without symptoms seemed to fare better?

260 Those without symptoms seemed to fare better psychologically, assuming “asymptomatic” to be “less dangerous”.

6. PLOS authors have the option to publish the peer review history of their article (what does this mean?). If published, this will include your full peer review and any attached files.

Reviewer #1: **Yes: **Nicola Jane Frances Pease

---

## [Author Response · Author response to Decision Letter 0]

12 Sep 2022

Dear Vanessa Carels,

Thank you for the opportunity to respond to the editor comments. We have addressed them point by point as described below. We think this has improved the quality of the review paper.

https://journals.plos.org/plosone/s/file?id=ba62/PLOSOne_formatting_sample_title_authors_affiliations.pdf".

Thank you. We have checked all the requirements and made the amended the files names.

Thank you. We have checked the reference list and made the necessary amendments.

Reviewer #1: An interesting piece of qualitative research, which adds to the current body of literature. The article is well structured, appropriate methodology, outcomes supported by appropriate quotes. On the whole the article is easy to read and follow. I do however have a few suggestions questions which I hope once addressed will make the article easier to follow.

Thank you.

• Paragraph below (page 8) does not read well, difficult to make sense of it.

137 Although the diagnosis of pulmonary embolism was an incidental finding discovered on a CT scan ordered for cancer staging or follow up, participants reported having thrombosis (PE/DVT) related symptoms even though these had not been recognised. where carers were present and willing to participate, dyad interviews were conducted.

Thank you for pointing out this omission. We have amended this omission.

• Re quote below – this may be correct but need to check did the respondent say ‘wrap gloves’

198 “Just need to be careful, if I am going anywhere, and I wrap gloves if I work in the garden.”

Thank you for pointing out this omission. This now has been amended.

• What information do you present to support this statement below - in capturing information on the 11 patients interviewed did you ask about symptoms of PE? To enable you to state those without symptoms seemed to fare better?

260 Those without symptoms seemed to fare better psychologically, assuming “asymptomatic” to be “less dangerous”.

Thank you for raising this question. Patients were asked about their life with cancer before the diagnosis of IPE, which includes any new or worsening symptoms. We have this information mentioned in the analysis, Theme one: 137-141 “highlighted”. As well in Theme two: 175, 176 “highlighted”.

Thank you.

---

## [Decision Letter · Decision Letter 1]

2 Oct 2022

PONE-D-22-00962R1Cancer patients’ experiences of the diagnosis and treatment of incidental pulmonary embolismPLOS ONE

Dear Dr. Ben Elhaj,

Thank you for submitting your manuscript to PLOS ONE. After careful consideration, we feel that it has merit but does not fully meet PLOS ONE’s publication criteria as it currently stands. Therefore, we invite you to submit a revised version of the manuscript that addresses the points raised during the review process.

We look forward to receiving your revised manuscript.

Kind regards,

Yoshihiro Fukumoto

Academic Editor

PLOS ONE

Reviewers' comments:

Reviewer's Responses to Questions

**Comments to the Author**

1. If the authors have adequately addressed your comments raised in a previous round of review and you feel that this manuscript is now acceptable for publication, you may indicate that here to bypass the “Comments to the Author” section, enter your conflict of interest statement in the “Confidential to Editor” section, and submit your "Accept" recommendation.

Reviewer #2: All comments have been addressed

Reviewer #3: (No Response)

2. Is the manuscript technically sound, and do the data support the conclusions?

Reviewer #2: Yes

Reviewer #3: Yes

3. Has the statistical analysis been performed appropriately and rigorously? 

Reviewer #2: I Don't Know

Reviewer #3: N/A

4. Have the authors made all data underlying the findings in their manuscript fully available?

Reviewer #2: Yes

Reviewer #3: Yes

5. Is the manuscript presented in an intelligible fashion and written in standard English?

Reviewer #2: Yes

Reviewer #3: Yes

6. Review Comments to the Author

Reviewer #2: This study was a qualitative research regarding with cancer patients’ experience of the diagnosis and treatment of IPE. Although this study addresses interesting issue, there are some drawbacks in the manuscript as described below;

1. This study was a single center study with a small number of patients. How have the authors estimate the required number of subjects for this qualitative research?

2. Why did the authors choose qualitative study to explore cancer patients’ experience of the diagnosis and treatment of IPE? Please mention the reason it was difficult to explore this issue by quantitative research.

Reviewer #3: First of all, the current new Reviewer (for revision) would like to congratulate the authors for the current interesting report. The authors seemed to revise the manuscript appropriately according to the previous initial review. The Reviewer do not have a major comment and would like to raise a few minor comments for hope to be helpful for the authors before publication.

Minor comments)

Title: The current study was “a qualitative study”, not a quantitative study. For general readers, it could be helpful to describe “a qualitative study” in the title.

Results: In the study period, how many potential patients were there? (All IPE patients) 13? If possible, please add the information, because this information could show readers the absence or presence of selection bias of patients.

In Conclusions section, the authors described that “Routine education for patients and clinicians should be provided.” However, as a clear message for general readers, routine education seemed to be vague description. What did the authors would like to intend?

7. PLOS authors have the option to publish the peer review history of their article (what does this mean?). If published, this will include your full peer review and any attached files.

Reviewer #2: No

Reviewer #3: **Yes: **Yugo Yamashita

---

## [Author Response · Author response to Decision Letter 1]

4 Oct 2022

Reviewer #2: This study was a qualitative research regarding with cancer patients’ experience of the diagnosis and treatment of IPE. Although this study addresses interesting issue, there are some drawbacks in the manuscript as described below;

1. This study was a single center study with a small number of patients. How have the authors estimate the required number of subjects for this qualitative research?

Thank you for raising this question.

This was a convenience sample of consenting participants completing an observational survey study. (Line 78/79 highlighted). We adopt the theoretical saturation concept in this qualitative study (the point at which the data collection process no longer offers any new or relevant data) (line 95 highlighted and referenced). This is because qualitative research methods are often concerned with an in-depth understanding of a phenomenon or are focused on meaning (and heterogeneities in meaning)—which are often centred on the how and why of a particular issue, process, situation, subculture, scene or set of social interactions.

2. Why did the authors choose qualitative study to explore cancer patients’ experience of the diagnosis and treatment of IPE? Please mention the reason it was difficult to explore this issue by quantitative research.

Thank you for raising this question.

The objective of this study was to explore cancer patients’ experience of the diagnosis of and living with incidental pulmonary embolism treated with anticoagulation. (Line 20/21 highlighted) and statistical procedures and numeric data were insufficient to capture how patients feel about their diagnosis and its effects on their daily life.

The aim of this qualitative research is to understand the social reality of individuals cultures as nearly as possible as its participants feel it or live it, rather than make statistical inferences. This involves asking participants about their experiences of things that happen in their lives. It enables researchers to obtain insights into what it feels like to be another person and to understand the world as another experience it, whereas quantitative research may not provide definitive answers to such complex questions.

Reviewer #3: First of all, the current new Reviewer (for revision) would like to congratulate the authors for the current interesting report. The authors seemed to revise the manuscript appropriately according to the previous initial review. The Reviewer do not have a major comment and would like to raise a few minor comments for hope to be helpful for the authors before publication.

Thank you.

Minor comments)

Title: The current study was “a qualitative study”, not a quantitative study. For general readers, it could be helpful to describe “a qualitative study” in the title.

Thank you for pointing out this omission. We have amended this omission. (Title highlighted) 

Results: In the study period, how many potential patients were there? (All IPE patients) 13? If possible, please add the information, because this information could show readers the absence or presence of selection bias of patients.

Thank you for raising this question.

Yes there were 13 potential patients. All been approached to participate. Two declined without giving any reason. (Line 79/80 highlighted).

In Conclusions section, the authors described that “Routine education for patients and clinicians should be provided.” However, as a clear message for general readers, routine education seemed to be vague description. What did the authors would like to intend?

Thank you for raising this question.

In the implications section of the paper, we referred to some potential ways that can be used to implement patients and health care education in patients’ care process, in the same way as patients are educated about neutropenic sepsis and spinal cord compression. (Line 312-317 highlighted).

However, we did not specify any potential actions of education as this need to be proved by future research, and this was out of the scope of this study.

---

## [Decision Letter · Decision Letter 2]

13 Oct 2022

Cancer patients’ experiences of the diagnosis and treatment of incidental pulmonary embolism. (A Qualitative Study)

PONE-D-22-00962R2

Dear Dr. Ben Elhaj,

We’re pleased to inform you that your manuscript has been judged scientifically suitable for publication and will be formally accepted for publication once it meets all outstanding technical requirements.

Kind regards,

Yoshihiro Fukumoto

Academic Editor

PLOS ONE

Additional Editor Comments (optional):

Reviewers' comments:

Reviewer's Responses to Questions

**Comments to the Author**

1. If the authors have adequately addressed your comments raised in a previous round of review and you feel that this manuscript is now acceptable for publication, you may indicate that here to bypass the “Comments to the Author” section, enter your conflict of interest statement in the “Confidential to Editor” section, and submit your "Accept" recommendation.

Reviewer #2: All comments have been addressed

Reviewer #3: All comments have been addressed

2. Is the manuscript technically sound, and do the data support the conclusions?

Reviewer #2: Yes

Reviewer #3: Yes

3. Has the statistical analysis been performed appropriately and rigorously? 

Reviewer #2: I Don't Know

Reviewer #3: N/A

4. Have the authors made all data underlying the findings in their manuscript fully available?

Reviewer #2: Yes

Reviewer #3: Yes

5. Is the manuscript presented in an intelligible fashion and written in standard English?

Reviewer #2: Yes

Reviewer #3: Yes

6. Review Comments to the Author

Reviewer #2: (No Response)

Reviewer #3: Thank you for your efforts of revision for the manuscript. The Reviewer do not have further comments. Congratulations for your important work.

7. PLOS authors have the option to publish the peer review history of their article (what does this mean?). If published, this will include your full peer review and any attached files.

Reviewer #2: No

Reviewer #3: **Yes: **Yugo Yamashita

---

## [Editor Report · Acceptance letter]

17 Oct 2022

PONE-D-22-00962R2 

Cancer patients’ experiences of the diagnosis and treatment of incidental pulmonary embolism (A Qualitative study)

Dear Dr. Benelhaj:

I'm pleased to inform you that your manuscript has been deemed suitable for publication in PLOS ONE. Congratulations! Your manuscript is now with our production department. 

Kind regards, 

on behalf of

Dr. Yoshihiro Fukumoto 

Academic Editor

PLOS ONE